# A Maximum Entropy Resolution to the Wine/Water Paradox

**DOI:** 10.3390/e25081242

**Published:** 2023-08-21

**Authors:** Michael C. Parker, Chris Jeynes

**Affiliations:** 1School of Computer Sciences & Electronic Engineering, University of Essex, Colchester CO4 3SQ, UK; 2Independent Researcher, Tredegar NP22 4LP, UK

**Keywords:** scale invariance, quantitative geometrical thermodynamics, Lagrange multipliers, Bayesian probability

## Abstract

The Principle of Indifference (‘PI’: the simplest non-informative prior in Bayesian probability) has been shown to lead to paradoxes since Bertrand (1889). Von Mises (1928) introduced the ‘Wine/Water Paradox’ as a resonant example of a ‘Bertrand paradox’, which has been presented as demonstrating that the PI must be rejected. We now resolve these paradoxes using a Maximum Entropy (MaxEnt) treatment of the PI that also includes information provided by Benford’s ‘Law of Anomalous Numbers’ (1938). We show that the PI should be understood to represent a family of informationally identical MaxEnt solutions, each solution being identified with its own explicitly justified boundary condition. In particular, our solution to the Wine/Water Paradox exploits Benford’s Law to construct a non-uniform distribution representing the universal constraint of scale invariance, which is a physical consequence of the Second Law of Thermodynamics.

## 1. Introduction

The ‘*Principle of Indifference’* (**PI**) is the intuitive principle that (in the absence of other evidence) an unbiased estimate of the outcome of a test will assign equal probabilities to the various possible outcomes. This appears to be a self-evident idea which seems to underpin all rational approaches to estimating probabilities. The idea of using ‘*Maximum Entropy*’ (**MaxEnt**) to determine equilibrium states of systems is also now well-established, as is (of course) Bayesian probability theory. It is, therefore, very disconcerting to find that there exist some circumstances in which the Principle of Indifference appears to be self-contradictory: if this is really the case, then a huge number of important results in all the sciences would be undermined.

Here, we will resolve a representative specimen of this set of paradoxes. We will show that for this set of ‘paradoxes’ (involving continuous possibilities rather than discrete ones), the PI entails (counter-intuitively) a *non-uniform* distribution representing the prior knowledge, nevertheless yielding a MaxEnt solution.

Benford’s Law [1] is the peculiar observation that in many real-life sets of numerical data, the leading digit is likely to be small. It was first observed by Simon Newcomb in 1881 [2], who commented: ‘*That the ten digits do not occur with equal frequency must be evident to anyone making much use of logarithm tables, and noticing how much faster the first pages wear out than the last ones*’. Although this is an expected statistical phenomenon [3], the reasons for it are remarkably obscure, and there remain a number of open problems [4]. Benford’s Law has since generated significant interest, including treatments that highlight its connections with entropy: Iafrate et al. (2015) [5] showed that the Law is derivable from a statistical mechanics treatment, with Don Lemons (2019) [6] extending their treatment to explicitly show the connection with thermodynamics.

It has become clear that Benford’s Law is associated with scale invariance, although Berger and Hill [4] give a simple counterexample for the (false) statement that ‘*To be Benford, a random variable or dataset needs to cover at least several orders of magnitude*’. Nevertheless, since we have demonstrated the validity of Quantitative Geometrical Thermodynamics (QGT) with an Euler–Lagrange variational calculus framework underpinning its Maximum Entropy (MaxEnt) approach to hyperbolic systems ranging over 35 orders of magnitude [7] (or more [8]), we expect such scale invariance to be present whenever the Second Law of Thermodynamics is at work. And we therefore also expect Benford’s Law, with its logarithmic character, to be ubiquitous (consistent with the fundamental and universal character of the Second Law) and indeed a ‘proxy’ for the Second Law complete with all the entailed physical limitations and constraints.

The Principle of Indifference is the simplest non-informative prior in Bayesian probability, mandating that, in the absence of any relevant evidence to the contrary, all possible outcomes should be treated as equally probable. ‘*The Principle of Indifference is a symmetry principle stating [that] logical symmetries should be reflected, in the absence of any discriminating information, in uniform* a priori *probability distributions*’ (Howson & Urbach, 2006 [9]).

However, Joseph Bertrand showed in 1889 [10] that the PI leads to apparent paradoxes for problems that have *infinite* sets of possible outcomes. Nicholas Shackel (2007) [11] analysed Bertrand’s Paradox (including Jaynes’ treatment of it [12]), concluding that in such cases, it continues to refute the PI. Of course, the first of the ‘plague of infinities’ of the PI is introduced at the outset since a uniform probability distribution entails a MaxEnt distribution characterised by an infinite temperature (since the temperature is inversely proportional to the Lagrange multiplier of the MaxEnt formulation: see *Technical Discussion* below).

The Wine/Water Paradox was introduced by Von Mises (1928) [13] as a resonant example of a ‘Bertrand paradox’ and has recently been re-analysed by Mikkelson [14], who concludes that the paradox is resolved if the symmetries of the problem are taken properly into account.

Briefly, an instance of the ‘Wine/Water Paradox’ is given by Mikkelson [14] as:

There is a certain quantity of liquid. All that we know about the liquid is that it is composed entirely of water and wine and that the ratio of wine:water (*w*) is between ⅓ and 3. So ⅓ ≤ *w* ≤ 3. Now, what is the probability that *w* ≤ 2?

Different ways of stating the Principle of Indifference give different answers to this question, and so the Wine/Water Paradox apparently represents an important counterexample to the PI, playing ‘*a curiously pivotal role in this discussion. Everyone seems to agree [that the Wine/Water Paradox] has no solution [and therefore that the PI] has fallen into serious disrepute among probability theorists*’ [14], even suggesting ‘*that the principle of indifference must be totally rejected*’ (Jaynes 1973 [12]; although he also says: ‘*the principle of indifference has been unjustly maligned in the past; what is needed was not blanket condemnation, but recognition of the proper way to apply it*’). Bas van Fraassen also thinks there is a fundamental failure of the PI: ‘*Probability is not uniquely assignable on the basis of a Principle of Indifference*’ [15].

In the language of Bayesian analysis, if a problem definition is to be considered complete and self-consistent, then it requires a complete specification of the *prior information* as well as the *data*, such that logical analysis from different points of view should lead to exactly the same solution.

Howson and Urbach [9] have stated that the PI ‘*is a symmetry principle*’, and associated with this property is Jaynes’ [12] idea of *transformation group theory* being applied to statistical problems invoking the PI; in particular, the assumption that changing the parameters (including the scale) of a problem should not change the state of knowledge. Such symmetry principles underlie both Special and General Relativity, where the hyperbolic rotations associated with Minkowski space-time form such a transformation group.

There are curious parallels with the statistical mechanical calculation of the thermodynamic entropy of a physical system, which depends on the granularity *chosen* to analyse the system under consideration. Elements of the system smaller than the graining represent the *microstates* of the system, which can be ignored since their permutations do not change the value of the entropy calculated. It is the *macrostates* of the system that represent the observational structure of the system.

In any case, when discussing the ignorance of a system, it is the Principle of Maximum Entropy (MaxEnt) that is important, particularly when a system is considered to be *under*determined. From a thermodynamical perspective, this is equivalent to the associated Lagrange multipliers being assigned a value of zero, which should be recognised as equivalent to assuming an infinite system temperature (as briefly mentioned above and elaborated in the *Technical Discussion* below). Such an assumption is both unphysical and also unjustified per se on informational grounds (since if we are completely ignorant of the temperature, we cannot assume any definite value). The MaxEnt principle is applied to systems associated with the *minimum* of information. Howson and Urbach [9] make a clear statement to this effect: ‘*Jaynes’s [MaxEnt treatment] appeals…to the criterion…of minimum information: …the least information… or… the fewest assumptions*…’ As Jaynes himself puts it, ‘*How do we find the prior representing ‘complete ignorance’? … the maximum entropy principle will lead to a definite, parameter-independent method of setting up prior distributions [such that] we express complete ignorance by assigning a uniform prior probability density*’ [12].

We regard this unjustified implicit assumption of infinite system temperature as being at the root of the disrepute of the PI and intend to approach the problem rather differently, invoking Benford’s Law as an explicit ‘proxy’ for a more physical application of the Second Law allowing the relevant MaxEnt parameters (including the Lagrange multiplier, but see *Technical Discussion*) to be properly determined and entailing physically consistent solutions.

It is interesting that Benford’s Law has not (to our knowledge) yet been applied to this class of problems. It is curious that Jaynes, who did so much to propose and support the principle of Maximum Entropy, neither saw the contradiction of effectively employing a specific (infinite) temperature for the assumption of a uniform prior probability density nor exploited Benford’s Law (which he was clearly aware of, citing it multiple times in his book on probability theory [16]).

## 2. Resolution of the Wine/Water Paradox

We want to know what is the median ratio *W* of the wine:water volume ratios *v/u*, *v′/u′* (choosing two representative ratios from the distribution), where for convenience (and without any loss of generality) we assume *v, v′*, *u* and *u′* are appropriately integer quantities, such that there is an equal probability of the wine:water ratios being above and below that median point. The key issue here is that an equivalent answer must be obtained for the symmetrical problem expressed using the inverse water:wine ratios, *u/v* and *u′/v′*. For convenience, we assume *v′/u′* > *v/u,* but of course, the inverse (symmetrical) assumption *v′/u′* < *v/u* may also be made. The wine:water ratio *w* can therefore be placed between the following limits:*v/u* ≤ *w* ≤ *v′/u′*(1a)

If the original statement of the problem does not employ integer values in Equation (1a) for the limits of *w* (such as in Mikkelson’s example), then it can be transformed by a common multiplication factor so that the limits are integers. Multiplying the wine:water ratio *w* by *uu*′, so as to define a transformed variable *x* = *wuu*′, then the limits of the scaled variable *x* (which no longer represents a ratio quantity) are given by:*u′v* ≤ *x* ≤ *uv′*(1b)

For any number system of base *B*, Benford’s Law states that the leading digit *N* for any number represented in that base *B* has a relative probability *p*(*N*) of occurrence of:(2a)$$p\left(N\right)=\log_{B}\left(1+\frac{1}{N}\right)$$

Since we also need to analyse the reciprocal quantities, we choose our base *B* as the product:(2b)$$B=\left(u+1\right)\left(u^{\prime }+1\right)\left(v+1\right)\left(v^{\prime }+1\right)$$

The limit quantities *P ≡ u*′*v* and *Q ≡ uv*′ (seen, in effect, in Equation (1b)) may be taken without any loss of generality to be, respectively, their own leading digit representation in the base *B*, so that using Benford’s Law, the natural probabilities of occurrence of *P* and *Q* are
(3a)$$p\left(P\right)=\log_{B}\left(1+\frac{1}{P}\right)$$
(3b)$$p\left(Q\right)=\log_{B}\left(1+\frac{1}{Q}\right)$$

We can immediately write the ratio of these two probabilities as
(4)$$\frac{p\left(P\right)}{p\left(Q\right)}=\frac{\log_{B}\left(1+1/P\right)}{\log_{B}\left(1+1/Q\right)}=\frac{\ln\left(1+1/P\right)}{\ln\left(1+1/Q\right)}$$
where, for convenience, we employ the natural logarithm. The most basic MaxEnt distribution (that is, the probability distribution with the fewest possible extraneous assumptions or constraints) is the negative exponential distribution so that the MaxEnt probability distribution for the scaled variable *x* is
*p*(*x*) = *A*exp(−λ*x*)(5a)
where λ is indistinguishable from a Lagrange multiplier. That is to say, the parameter λ can be considered here to represent the physical constraint of scale invariance that has been introduced into the problem formulation by the ubiquitous influence of the Second Law of Thermodynamics. For *x* lying between the two numbers *P* and *Q* (as per Equation (1b)),
(5b)$$\int_{P}^{Q}Ae^{-\lambda x}dx=1$$
hence:(5c)$$A=\frac{\lambda }{e^{-\lambda P}-e^{-\lambda Q}}$$

It is clear that the MaxEnt distribution *A*exp(−λ*x*) is *under*-determined since it has two variables, *A* and λ, but only one constraint (Equation (5b)). This under-determination for *A*exp(−λ*x*) can be thought to have generated the Wine/Water Paradox since a unique designation for each of *A* and λ was not available. But, asserting Benford’s Law and recasting Equation (4) with the help of Equation (5a):(6)$$\frac{p\left(P\right)}{p\left(Q\right)}=\frac{\exp\left(-\lambda P\right)}{\exp\left(-\lambda Q\right)}=\frac{\ln\left(1+1/P\right)}{\ln\left(1+1/Q\right)}$$

Thus, Equation (6) represents a new independent relation allowing for the unique determination of the exponential parameter λ, and therefore the most likely (MaxEnt) distribution for the parameter *x*, since Equation (5c) can then be used to uniquely determine *A*. Note that it is clear that λ ≠ 0 unless *P* = *Q*; that is to say, only the trivial case (and, in effect, a null wine/water proposition) leads to what might be considered the uniform probability distribution (with λ = 0) conventionally associated with the PI.

Thus, the initial aspect of von Mises’ conundrum can now be straightforwardly solved; that is to say, the value for the median probability is given by the value *x* = *X*, where *X* (with P≤X≤Q) is uniquely determined by the following condition:(7a)$$\int_{P}^{X}Ae^{-\lambda x}dx=\int_{X}^{Q}Ae^{-\lambda x}dx=\frac{1}{2}$$
using the LHS of Equation (7a) leads to a closed solution for *X*:(7b)$$X=\frac{-1}{\lambda }\ln\left(e^{-\lambda P}-\frac{\lambda }{2A}\right)$$

Transforming back into the ratio *w* leads to a median wine:water ratio given by *W* = *X*/*uu*′:(7c)$$W=\frac{-1}{uu^{\prime }\lambda }\ln\left(e^{-\lambda P}-\frac{\lambda }{2A}\right)\;$$

Considering now the *reciprocal* ratios, and in particular the reciprocal variable *y* (≡1/*w*), then the relative proportions of water and wine are now considered to be *u/v* and *u′/v′*, such that for *u/v* > *u′/v′,* we have:*u′/v′* ≤ *y* ≤ *u/v*(8a)

Then, as before, we may, without any loss of generality, multiply the appropriate relative inverse ratios *u/v* and *u′/v′*, by the factor *vv′* so as to ensure integer quantities obeying Benford’s law as per Equation (3), and thereby consider the scaled variable *z* ≡ *vv′y*, such that (assuming again a MaxEnt probability distribution for *z*)
*u′v* ≤ *z* ≤ *uv′*
(8b)
or indeed
*P* ≤ *z* ≤ *Q*
(8c)
with
*p*(*z*) = *C*exp(−Λ*z*)(8d)

With the resulting limit quantities *P ≡ u*′*v* and *Q ≡ uv*′ still applying and the MaxEnt equation Equation (8d) equivalent to Equation (5a), it is also now clear that *C* ≡ *A* and Λ ≡ λ, such that it is also clear that the median point corresponds to the same equation Equation (7b). That is, the problem has the same solution (as required) as its inverse (*mutatis mutandis*).

## 3. Technical Discussion

In applying this theory to the Wine/Water Paradox using Mikkelson’s parameters (1 ≤ *x* ≤ 9, transferring to integer numbers simply by multiplying Mikkelson’s range by a factor 3), we now have the means to find the value of *W* such that there is an equal probability of the actual ratio being above or below *W*. The above analysis reveals that the solution to the Water–Wine Paradox is given by (solving Equation (6))
λ = 0.235481801(9a)
with the associated value for *A* given by
*A* = 0.35142409(9b)
and a median value *W* found from Equation (7c):*W* = 1.11420745 (9c)

This median value of the probability distribution represents the ratio of wine/water such that there is an equal probability of the actual ratio being above or below that value. As required, the inverse problem has the same median value.

It is of interest to note that the median probability point associated with *W* is not unity (that is, equal quantities of water and wine). This result is an interesting aspect of the wine/water problem that invites some comment. In particular, we note that in our integer terms of *P* and *Q*, then Mikkelson’s parameters are equivalent to the range lying between 1 and 9, with a mid-point of 3, corresponding to the ‘geometric’ mid-point. Clearly, the general result that *W* be unity would, in turn, imply that the solution for *W* is the geometric mean, which is attractive since it appears scaleless. However, the geometric mean does not actually conform to the logarithmic (hyperbolic) nature of the universe, as exemplified by the entropic basis of the Second Law (see Eq.1b of Parker & Jeynes 2019 [17]). Equation (1) shows that Benford’s Law is also ‘logarithmic’ in the same way, which is why we called it a ‘proxy’ for the Second Law. When considering the median probability from the perspective of physical quantities, the logarithmic calculation for *W* (as per Equation (7c)) is the more meaningful physical approach.

Although not exactly the same, the velocity addition rule of Special Relativity offers a related (hyperbolic) means to combine two velocities. Similarly, the addition of optical Fresnel-based reflection amplitudes (with phase properties) for the overall probability amplitude of the reflection from a multilayer dielectric stack also follows a hyperbolic tangent (*tanh*) addition formalism (see Corzine et al. [18]). That is, neither a geometric nor an arithmetic addition occurs in either of these cases. As is well-known, the science of probability has always tended to defy intuition by offering unexpected and surprising solutions: the Monty Hall problem (see Enßlin et al. 2019 [19]; Enßlin & Westerkamp 2019 [20]) is just one example of many; Edwin Jaynes also delighted in exploiting the Maximum Entropy machinery to objectively solve problems (such as the loaded dice; see Jaynes 1978 [21], Jaynes 1982 [22]) in a manner aimed at disconcerting those unfamiliar with these methods.

Jeffrey Mikkelson [14] considers that he has ‘dissolved’ the Wine/Water Paradox by ‘epistemically’ distinguishing between ‘primary’ and ‘derivative’ facts. We have shown that explicitly invoking Benford’s Law yields a more clear-cut (and satisfactory) resolution.

Marc Burock [23] does not like Mikkelson’s solution since he regards it as silently introducing extra information. Instead, he draws attention to ‘*the joint sample space of a ratio and its inverse*’ and claims that applying the PI to this space resolves the paradox. In our opinion, this can be regarded as a sort of scale invariance that we implement *explicitly* using Benford’s Law.

For the question ‘What is the probability *P*^*^ that *w* ≤ 2?’, we again multiply Mikkelson’s parameters by three and calculate the integrated probability for *P* ≤ *x* ≤ 6, with *P* = 1:(10)$$P^{*}=\overset{6}{\underset{P}{\int }}Ae^{-\lambda x}dx=\frac{-A}{\lambda }{\left[e^{-\lambda x}\right]}_{P}^{6}=\frac{-Ae^{-\lambda P}}{\lambda }\left\{e^{-\left(6-P\right)\lambda }-1\right\}=0.81595117$$

Michael Deakin published a sophisticated discussion [24] of both Mikkelson’s and Burock’s conclusions. He points out that Mikkelson finds *P** = 5/6 = 0.833, whereas Burock finds *P** = 0.764 (from Equation (10) above, we find *P** = 0.816). He concludes that the problem as posed may have any solution in the interval ½ ≤ *P** ≤ 1. We regard the problem as rather better-posed than he thinks it is, with a definite solution supplied by the extra information intrinsic to Benford’s Law. Gerville-Réache [25] also discusses both Mikkelson’s and Burock’s treatment, concluding (with Deakin) that the problem as posed admits multiple solutions (including his preferred *P** = 15/16 = 0.9375).

John Norton comments, very plausibly, ‘If our initial ignorance is sufficiently great, there are so many ways to be indifferent that that the resulting equalities contradict the additivity of the probability calculus. We can properly assign equal probabilities in a prior probability distribution only if our ignorance is not complete and we know enough to be able to identify which is the right partition of the outcome space over which to exercise indifference’ [26]. His is a paper on ‘probabilistic epistemology’, but here we prefer to avoid epistemological questions in favour of explicit physics (although it is not always possible to avoid metaphysics [27]).

We have used Benford’s Law to generate a non-uniform (but still MaxEnt) prior to our fully Bayesian inference, resolving the ‘Paradox’. One might ask (and thanks to another anonymous referee for pointing this out) whether other priors might also be used. Simkin and Roychowdhury (2011) [28] describe a variety of other ‘*power law distributions*’ (such as the heuristic Zipfian distribution) which show similar scale-independent behaviours. It turns out that these are all related both to each other and to Shannon’s *information entropy*. We have chosen to use Benford’s Law since it is sufficiently general and well-founded statistically.

We have above repeatedly alluded to the issue of Lagrange multipliers, an issue which highlights an apparently unphysical aspect of the most basic form of the *Principle of Indifference* (PI). The point here is that the Maximum Entropy method looks for stationary solutions of the Lagrangian system given the constraints. These constraints may be represented by the (constant scalar) ‘Lagrange multipliers’. It is a standard result (see, for example, Equation (5.38) in Caticha 2008 [29]) that, for the Boltzmann distribution, the Lagrange multiplier representing the energy conservation constraint is inversely proportional to the temperature, and our Equation (5a) features the parameter λ that is indistinguishable from such a Lagrange multiplier.

It is also obvious that for uniform distributions (as are usually implied by the PI: for such systems, the relevant probability distributions are independent of particular constraints), the Lagrange multipliers must be zero. This, in turn, implies an (unphysical) infinite system temperature. Here, we draw attention to this problem while using Benford’s Law to select one of the families of Maximum Entropy solutions, one with a parameter that is equivalent to a non-zero Lagrange multiplier. This problem has a solution *not* given by a uniformly distributed probability function, such as one would intuitively expect from the PI. Perhaps this has been such a persistent Paradox precisely because the simplest form of the Principle of Indifference as applied to the Wine/Water Paradox entails a trivial solution (*P* = *Q*) even though the probability functions of sensible solutions are not distributed uniformly, even as they remain MaxEnt solutions.

It is worth mentioning (with thanks to the anonymous referee who pointed this out) that the standard way of measuring the entropy *S* of a system is to measure its chemical potential μ and then use the appropriate Maxwell relation (∂*S*/∂*n*)*_T_* = (∂μ/∂*T*)*_n_*, where the number of particles *n* in the system is assumed to be very large so that the infinitesimals are approximated reasonably well. In this way, one can obtain an ‘*entropy per particle*’ for the system [30]. But again, to obtain the chemical potential, one usually uses a Lagrange multiplier method and sets it to zero (to effect the ‘equilibrium’ condition). But as Baerends (2022) [31] insists, ‘*The crux of the Lagrange multiplier method is that it allows one to use full derivatives. This requires that the full derivative is defined* …’, which, strictly speaking, it is not. This is another way of emphasising that the methods we use assume the validity of the differential calculus, which for finite systems, can be only an approximation.

## 4. Philosophical Discussion

The question raised by anonymous referees is, ‘*what is the claimed practical consequence of this result?*’ Practically, it gives us a new security in the ***validity of the Principle of Indifference***! The Water/Wine Paradox (and other related paradoxes) have plagued the analytical treatment of logical inference for over a century now. So, being assured of the resolution of this disconcerting question lurking at the very foundations of our methods of logical inference must be a significant advance. After all, we need to know that logical inference really is as logical as it is claimed to be. Physicists often use mathematical methods rather casually, but in the end, they always want to know that the mathematicians have correctly sorted out the details.

Since we have invoked the Second Law of Thermodynamics repeatedly, another question raised is, ‘*what specifically have these results to do with the Second Law?*’ The Maximum Entropy techniques ubiquitous today presuppose the Second Law, and our resolution of the Paradox is demonstrably MaxEnt (and therefore depends fundamentally on the Second Law).

That is to say, the Wine/Water situation poses a rather clearly defined estimation problem which has been widely discussed. Up to now, it has seemed that there was no clear solution to this estimation problem that did not stray into metaphysical treatments. But we have shown that metaphysics is not required, only the explicit introduction of an extra prior, a prior that is clearly and universally valid. This prior (Benford’s Law) is actually a mathematical statement of a general truth about any numerical representation of things so that it is logically necessary (and always applicable): it is not a contingent (physical) truth. The reason that it is so powerful in our application is that it may be validly thought of as a proxy for *scale invariance*, which expresses the (relativistic) idea that a universal truth should be valid at any scale. We assert that thermodynamics is also scale-invariant, and we have physically demonstrated this invariance by showing that our thermodynamics is valid over at least 35 orders of magnitude, from the subatomic to the cosmic [7]. That is, Benford’s Law can also be thought of as a proxy not only for scale invariance but also for priors required by thermodynamics itself.

A further question raised is, ‘*under what conditions has this result empirical meaning?*’ Could we perform an experiment and measure these probabilities? Yes, we could. But it would be as pointless to ‘confirm’ the result ‘experimentally’ as it would be to ‘confirm’ the equiprobability of heads/tails in a coin-toss ‘experiment’. Can the rules of logical inference be confirmed ‘experimentally’? No, they cannot. To interpret the meaning of any experiment, the rules of logical inference must be assumed. The metrologists have made this very clear. The *Guide to the Expression of Uncertainty in Measurement* (the ‘GUM’, published under the aegis of the BIPM [32]) states, ‘*A measurand is in many cases not measured directly, but is indirectly determined from other quantities to which it is related by a measurement model. The measurement model is a mathematical expression (or a set of such expressions), comprising all the quantities known to be involved in a measurement. It enables a value of the measurand to be provided and an associated standard uncertainty to be evaluated*’. The GUM explicitly depends on the VIM (the *International Vocabulary of Metrology* [33]), which carefully states, ‘*The objective of measurement is not to determine a “true value” as closely as possible. Rather, it is assumed that the information from measurement only permits assignment of an interval of reasonable values to the measurand, based on the assumption that no mistakes have been made in performing the measurement*’. Note that the empirical fact permanently confronting metrologists is that uncertainty is unavoidable. What conclusions can be validly drawn in the ubiquitous presence of uncertainty?

One does not usually associate metrology with philosophical ideas like ontology and epistemology; nevertheless, Mari et al. (2013) [34] are metrologists who have drawn a careful philosophical distinction between being a quantity and being measurable. They point out that this distinction is an *ontological* one and, moreover, that ‘*measurement is primarily an epistemic process*’. It should be recognised that the very idea of ‘empirical meaning’ is itself heavily and irreducibly philosophical.

## 5. Conclusions

We have shown that the value of λ obtained explicitly (Equation (6)) resolves the Water/Wine Paradox, being based on a more physically realistic expression of the *Principle of Indifference*, and which remains valid even when the problem is expressed in a different (but symmetrical) manner.

The ‘paradox’ is one famous example of a class of paradoxes described by Bertrand, and its resolution here for one case is expected to resolve the other cases, too, *mutatis mutandis*.

We regard the ‘paradox’ as appearing paradoxical because it is ostensibly under-determined, as stated, so that different solutions seem valid for an apparently well-posed problem. This underdetermination is an expression of unrecognised and unstated priors disturbing the analysis, while it is, of course, well-known that a correct Bayesian analysis must also correctly state all the prior knowledge of the system.

We solve the ‘paradox’ by explicitly supplying the missing prior (in the form of Benford’s Law): namely, the condition of scale invariance. Other commentators have also noticed this prior but have treated it metaphysically. Here, we treat it physically.

Our treatment shows that the Principle of Indifference does not necessarily imply the uniform probability distribution one usually expects. This is because a uniform distribution implies a null Lagrange multiplier, which in turn implies an (unphysical) infinite system temperature. But it is important to note that a null Lagrange multiplier also implies *independence* of relevant constraints; this independence has been (illegitimately) smuggled in as a further (unacknowledged) implicit assumption. We have shown rigorously that the explicit assumption of scale invariance required by the Second Law and implemented using Benford’s Law allows a distinct and consistent Maximum Entropy solution to the wine/water problem with a non-zero Lagrange multiplier explicitly evaluated.

This has wider importance because the *Principle of Indifference* underlies all Maximum Entropy (Bayesian) analysis, and the suggestion that this Principle is invalid in general (with specific supposed counterexamples) undermines the usefulness of the Maximum Entropy methods that are now ubiquitous. But we have constructively demonstrated the validity of this Principle by explicitly resolving the (alleged) Paradox.

## Data Availability

No new data were created or analyzed in this study. Data sharing is not applicable to this article.

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
