# Peer review of "A Maximum Entropy Resolution to the Wine/Water Paradox"

_entropy, 2023, doi:10.3390/e25081242_

Round 1

Reviewer 1 Report

Line 18: upon initial reading, it occurred to me that Benford’s law must be a consequence of logarithmically distributed probabilities. The discussion on lines 176-185 seem to confirm this interpretation? I would find it useful to clarify this idea at the outset.

General comment: The authors discuss the situation where a “Lagrange multiplier” (inverse temperature) is set to zero, corresponding the case of infinite temperature which is argued to be nonphysical. A physical situation where another Lagrange multiplier is set to zero to obtain thermal equilibrium involves the chemical potential. This mu=0 is used when the number of particles (N) is not fixed, such as for phonons and photons. Perhaps this would be another useful example and analogy for the authors to use.

Reviewer 2 Report

1. References styles varied too much.  

2. Authors favor run-on sentences, or sentences with too many semi-colons (see my markup).

3. Equations 5.b and 5 .c should have an explanation added (footnotes) for novice readers and a resource cited. 

Reviewer 3 Report

The authors consider the wine/water paradox in probability and resolve it by enforcing Benford's law. 

In my opinion the paper does not match the quality standard of the journal Entropy. In particular, it seems mostly focused on a very specific problem in probability, which is solved assuming a certain form of the prior in Bayesian probability. Moreover, I do not see any relation with more general physical or mathematical problems. The connection with the second law of thermodynamics is not clear.

I suggest the authors to submit the paper to a more specialized journal.

Reviewer 4 Report

The authors present a very educative paper with a lot of thinking in various directions. I recommend publication in principle but see some difficulties in the used distribution which could maybe discussed. The assumption of finite temperature or Lagrangian parameter is fine since any infinitesimal small one would lead to the same expressions. Besides normalization, the second condition to determine the distribution is chosen as Benfords law. Here it is to ask why relying on this law and not e.g. on a Zipfian distribution. In fact the answer could be given in the presented formalism for any law and the discussion maybe enriched by pointing out that the results depends on this underlying assumption. In this sense I cannot share the optimism that the paradox is solved. Nevertheless, the problem is stated and possible solutions dependent on the chosen distribution are worked out. This alone justifies publication.

Round 2

Reviewer 2 Report

approved for publication without changes

approved for publication without changes

Author Response

Thanks to the Reviewer

Reviewer 3 Report

I am not satisfied by the answer from the authors. They only added some general sentences in the Introduction and in the Conclusion, without providing a more specific discussion of the relevance of their results. Moreover, I asked for a clarification of the connection with the second law of thermodynamics, which is mentioned several times in throughout the paper, but not further explanation has been illustrated. 

Round 3

Reviewer 3 Report

The authors have explained in more detail some the points addressed in the paper. I suggest publication in the presente form.